# Continuous Professional Development for Public Sector Pharmacists in South Africa: A Case Study of Mapping Competencies in a Pharmacists’ Preceptor Programme

**DOI:** 10.3390/pharmacy8020096

**Published:** 2020-06-03

**Authors:** Mea van Huyssteen, Angeni Bheekie, Sunitha C Srinivas, Azeezah Essack

**Affiliations:** 1School of Pharmacy, University of the Western Cape, Private bag X17, Bellville, Cape Town 7535, South Africa; abheekie@uwc.ac.za (A.B.); 3034735@myuwc.ac.za (A.E.); 2Faculty of Pharmacy, Rhodes University, Grahamstown 6140, South Africa; s.srinivas@ru.ac.za

**Keywords:** collaborative practice, competency-based, continuous professional development (CPD), pharmacy education, pharmacy practice, preceptor

## Abstract

Lifelong learning among healthcare practitioners is crucial to keep abreast of advances in therapeutic and service delivery approaches. In South Africa, continuous professional development (CPD) was mandated (2019) for re-registration of pharmacists to illustrate their learning according to the South African Pharmacy Council’s (SAPC) competency standards. This paper uses a preceptor programme linked to the University of the Western Cape School of Pharmacy’s service learning programme to map the competencies employed by pharmacist preceptors in primary care public healthcare facilities in Cape Town in an attempt to encourage completion of their annual CPDs and strengthening the academic-service partnership. Competencies identified were divided into input competencies related to the preceptor’s role in designing and implementing the educational programme in their facilities and assisting students to complete their prescribed learning activities, and output/outcome competencies that emerged from preceptors identifying the facility needs and employing their input competencies. Input competencies pertained to education, leadership, patient counselling, collaborative practice and human resources management. Output competencies related to pharmaceutical infrastructure, quality assurance, professional and health advocacy, primary healthcare, self-management and patient-centred care. The preceptor programme enabled pharmacist preceptors to employ several competencies that are aligned with the SAPC’s competency framework.

## 1. Introduction

Lifelong learning among healthcare practitioners is important to keep pace with rapid transformations which inform the practice environment [1], ensure patient safety and quality of care [2]. Two approaches that have been used to ensure and promote lifelong learning globally in pharmacy include continuing professional development (CPD) and continuing (pharmacy/medical) education (CE) [3]. In their CPD/CE global report (2014), the International Pharmaceutical Federation (FIP) reported that CPD/CE was required to maintain registration in 33 out of 66 countries that were investigated [3]. At the time of publishing, South Africa was in the process of developing an “advanced practice” competency framework [3].

The advanced practice framework, i.e., the South African competency standards for pharmacists was revised in 2017 to align with the FIPs global competency framework [4], and contextualized to South African resources, needs, and policies that reframe the health system to implement universal health coverage which included: the National Development Plan [5], National Health Insurance (NHI) [6], and National Core Standards [7]. The introduction of the new competency standards was the driver of the development of the new *Regulations relating to continuing professional development, 2019 (BN:42464)* [8] that were published for implementation on 17 May 2019 in terms of section 33(1)(o) of the Pharmacy Act, 1974 (Act No. 53 of 1974) of South Africa [9]. Until 2019, the South African Pharmacy Council (SAPC) listed CPD for pharmacists as a “professional obligation” rather than a mandatory requirement [10]. The new regulations make the annual completion of six CPD entries mandatory for annual re-registration. In comparison, most other health professions in South Africa use a CE-based credit system that requires the accumulation of a specified number of hours of training for the re-registration of health practitioners [11]. Subsequently, most lifelong learning activities which are available in South Africa have been designed and accredited for a CE system. 

CE tends to focus on participation in education and training events. In comparison, CPD is a cyclical process that requires reflection on practice (i.e., identifying knowledge/practice needs), planning of the educational activity, implementing the training (attending the training) and applying the acquired knowledge to practice and assessing its effectiveness. The completing of a CPD event thus incorporates more elements than CE [12]. Rouse [13] stated that the three important features of CPD were “practitioner-centred and self-directed, practice related and outcomes orientated” [13] (P. 2074).

Although there is ample evidence to support the effectiveness of CE in terms of knowledge, performance and health outcomes, evidence on the effectiveness of CE in medical education has showed greater improvements in performance and health outcomes with CE activities that were interactive, used different methods of presentation, employed multiple training exposures over a prolonged period of time, and was focussed on outcomes considered to be important to practitioners [14]. In the pharmacy, this evidence seems to be reflected in the American Council for Pharmacy Education’s (ACPE) classification of continuing education (CE) activities to distinguish between: knowledge-based, application-based and practice-based activities [1]. Similarly, Australia recently revised their CPD framework to assign more credits for practice improvement activities (three credits per hour) than knowledge-based activities (one credit per hour) [15].

Since CPD tends to focus on practice needs, there seems to be a greater return on investment compared to CE [12,16]. A 10 month randomised controlled study that compared CPD with CE found that pharmacists who employed a CPD approach reported more improvement in patient care changes, professional knowledge, skills and attitudes than those who employed the traditional CE approach [17]. In addition, competency-based CPD [18] includes behavioural statements which cement the practice aspect of any CPD activity [12].

However, some of the problems that pharmacists associated with CPD is that completing CPD was more time consuming than CE, and no preferred model of presentation for CPD/CE was identified as most effective [12]. In addition, most of the outcomes of current CPD programmes were not aligned with expectations from employers, regulators and patients [1]. In addition, the monitoring of CPD may be resource intensive [19]. Taking into consideration all of the aforementioned problems, innovative solutions are needed for resource constrained settings such as South Africa, where the evidence for CPD is scarce, as most evidence is documented predominantly from developed countries that have mandated CPD [12]. The question that arises is: How could an already established practice-based programme be harnessed to enable and motivate pharmacists to complete their annual CPDs? 

Indeed, longitudinal CPD programmes that facilitate CPD activities over extended periods of time have been suggested as an external motivator for pharmacists to complete CPDs [20]. This paper uses a preceptor programme (linked to the University of the Western Cape School of Pharmacy’s service learning programme) to (1) identify competencies employed by preceptors (in primary care public healthcare facilities in Cape Town), and (2) map these competencies according to the new competency standards for pharmacists in South Africa. This information will be used to shape future engagement between university faculty and preceptors to increase the synergy between preceptors, faculty and students in participating in the programme, i.e., strengthening the reciprocity of the academic-service partnership, by aligning with a mandatory activity required by the SAPC as part of the programme. Indeed, CPD has been established as an essential competency for preceptors who mentor students [21]. In the rest of this paper, the completion of CPDs according to the SAPCs competency standards is explained. The preceptor programme and data collection is contextualized, and finally, competencies are mapped and discussed.

## 2. South African CPD Context

The first step of the CPD process includes the completion of an annual declaration that confirms the practicing/non-practicing status of the pharmacist. Next, each pharmacist indicates which competencies as stipulated in “Competency Standards for Pharmacists in South Africa” [22], they employ in their practice, as it is accepted that not all pharmacists would employ all competencies. CPD activities for the year must be submitted on the online portfolio system of the SAPC in accordance with the CPD cycle that follows the four-step process of reflection on practice, planning, implementation and evaluation or reflection on learning.

The competency standards are organized into clusters of competencies referred to as domains (Table 1). Each domain or competency cluster consists of six to nine different competencies. Each competency may consist of one or more behavioural statement that specifically describes behaviours associated with that competency. Behavioural statements are described according to three different levels of practice divided into entry level practice (year one to three of practice), intermediate practice (year three to seven of practice) and advanced practice (more than seven years of practice) [22]. An example includes the competency of (5.7) collaborative practice under the Domain (or competency cluster) of (5) professional and personal practice. This competency contains only one behavioural statement (5.7.1) which are described as (5.7.1.1) *Participate as a team member in a multidisciplinary team* (entry level practice), (5.7.1.2) *Practice in a multidisciplinary team with cognisance of the roles and services delivered by healthcare and other related professionals* (intermediate practice), and, (5.7.1.3) *Advocate for the inclusion of pharmacists in all multidisciplinary healthcare teams* (advanced practice) [22].

## 3. Case Study: Pharmacist Preceptor Programme

The School of Pharmacy at the University of the Western Cape (UWC) has been strengthening its Service Learning in Pharmacy (SLiP) programme and partnership since 2002 [23]. The primary service-learning goal is to ensure that graduates have the knowledge and skills to become population, community and patient-centred pharmacists who are committed to addressing South Africa’s pressing primary healthcare needs.

The service-learning programme consists of a tripartite partnership between the patients (or community), the university (faculty and students) and health service agencies. The initial and subsequently the longest standing partnership to date has been with public sector pharmacists practising under Cape Town’s Metro District Health Services (MDHS) and those from the Western Cape provincial pharmaceutical services health department. The School’s mandate for the service-learning partnership is to engage in an interdependent and reciprocal relationship that underpins the programme. As such, maintaining a working relationship with the MDHS facility-based pharmacists and the pharmaceutical services management is a priority. Subsequently, faculty staff have hosted annual facilitators’ workshops since 2011 for the facility-based pharmacists who directly facilitate pharmacy students at health facilities, pharmacists at management levels of the Department or Health, namely, the deputy-directors of pharmaceutical services at (MDHS) sub-district level and pharmacists representing the provincial pharmaceutical services. The purpose of these workshops was to introduce the service partners to: the service-learning programme’s objectives, student learning objectives, the designing of learning activities for their facility context, the standardisation of on-site student assessments, to provide guidance on good facilitation skills, update them on curricular changes, and provide student feedback and engage with facilitators on their experiences with learning activities. A memorandum of agreement was formalised between the university and MDHS (2016), which is currently being revised.

Since its inception, SLiP has operated primarily as a fourth year programme across public sector pharmacies (2002), but later, its expansion led to inclusion of additional undergraduate programmes, service partners and learning sites (Table 2). In 2015, SLiP transitioned from the old curriculum (pre-2013) where it was presented in the fourth study year, to a new curriculum (post-2013) where it was presented as a third year programme. In 2016, the new fourth year programme, the Patient Care Experience (PaCE), was launched, using the established service-learning sites. The SLiP programme thus formed the operational foundation along with its facilitators and pharmacy-based activities for the PaCE programme.

The pre-2013 SLiP programme consisted of two arms; the service learning arm, in which an on-site pharmacist (referred to as a facilitator) facilitated students in delivering pharmaceutical services from the pharmacy at various public healthcare facilities (including primary healthcare clinics and hospitals); and the clinical training arm, in which UWC faculty facilitated students in one or two public hospital wards without on-site pharmacist involvement. For service learning, students were required to complete three weeks at a designated health facility. The pharmacist facilitator’s role was to select and design the learning activities from the menu of options (Figure 1), engage the facility staff for the students, orientate and host students, implement these learning activities during SLiP, and assess the students’ pharmacotherapeutic dispensing skills through an Objective Structured Dispensing Examination (OSDE) and overall performance at the site, which included their ability to apply theoretical knowledge, identify and address challenges, follow instructions, and interact professionally with staff and patients (Figure 1).

The pre-2013 SLiP programme used the term “facilitator” to describe on-site pharmacists who supervised students, which is a term commonly used in service learning [24]. Yet, when the PaCE programme was introduced with its focus on clinical learning, the term “preceptor” was adopted due to its prevalent use in clinical pharmacy training. In pharmacy literature, the term “preceptor” is most commonly used to describe a pharmacist that is involved in the clinical training and supervision of pharmacy students at the experiential learning sites [25].

The PaCE programme, which was introduced in 2016, required students to attend a three-day orientation programme and participate in two rotations; five weeks at a primary care clinic or community health centre (CHC) and five weeks at a hospital facility. In contrast to the menu of options available for SLiP, the PaCE programme was more prescriptive in assignments, which included: the development of pharmaceutical care plans, medicine history taking and reconciliation, interprofessional education activity, the design of a patient education poster, the conduction of an in-service training session and the implementation of a quality improvement project (Figure 2). These assignments are described in detail elsewhere [26].

In terms of student supervision for the PaCE programme, both UWC and on-site preceptors participated. Although a UWC preceptor was responsible for student assessment at all their assigned sites, due to staff shortage, they only managed on-site supervision, bed-side teaching and assessment for students in the hospital rotation. The on-site preceptor’s roles were to engage the facility staff for the student rotation, orientate and introduce students to the designated facility staff, offer guidance to students in identifying assignment topics for in-service training, patient education poster and group projects, assist students with obtaining interprofessional clinical learning opportunities, facilitate engagement with patients for poster development, medicine history and reconciliation and access to clinical information required for pharmaceutical care plan development.

## 4. Method

Up until 2015, the research conducted into the service learning programme had focussed on general programme description [27] and student learning experiences [23]. The first data collection activities from service partners was part of a research project to align pharmacy education with social accountability (2015) that intended to evaluate longitudinal outcomes of the service learning programme and to use this evidence-based approach to improve the programme and strengthen the tripartite partnership. Baseline data were collected from preceptors in 2016 during an expansion phase of the service-learning curriculum and included preceptor workshops and individual interviews with champion preceptors. A second round of data were collected in 2019 from a primary healthcare preceptor feedback workshop and individual interviews with those preceptors who could not attend the workshop. Ethics approvals were obtained from the UWC Senate Research Ethics Committee (15/6/95) and from UWC Biomedical Research Ethics Committee (BM18/6/14). Additionally, the reflections of two faculty members (MH, AB) who precepted at hospital sites (2016 and 2017) and community sites (2018 and 2019) were used for this paper.

## 5. Results

The 2016 data was collected from 43 pharmacists participating in two workshops (n = 28), and n = 27 participants, respectively) and seven interviews (n = 8 participants). The participants represented CHCs (n = 20), hospitals (n = 9), sub-district management (n = 3) and the provincial pharmaceutical services office (n = 1). The 2019 data was collected from 21 pharmacists participating in one workshop (n = 13 participants) and seven interviews (n = 8 participants). The participants represented CHCs (n = 19), sub-district (n = 1) and the provincial pharmaceutical services office (n = 1). Twelve (12) participants participated in both 2016 and 2019 in one of the data collection activities. Longitudinal participation data of participants and the facilities they represented are summarised in Appendix A.

Competencies were mapped according to the domains and competencies reflected in the Competency standards for pharmacists in South Africa [22]. In addition, the behavioural statements are indicated in italics where possible with the number allocated in the competency standards for the reader’s reference. Preceptor competencies were divided into input competencies and output/outcome competencies that constituted the overarching themes (Figure 3). Input competencies correlated with the role of the preceptor in managing the facility’s preparation for students and student learning activities, orientating and facilitating students during their time at the facility, and enabling students to conduct and complete their prescribed assignments. Output/outcome competencies related to the role of the preceptor in identifying the facility’s needs relating to topics for the patient education poster, in-service training and quality improvement project assignment and those competencies that emerged as a consequence of the preceptor performing their assigned roles. As such, the input competencies represented four domains, namely (2) safe and rational use of medicines, (4) organisation and management skills, (5) professional and personal practice, (6) education, critical analysis and research, and the output/outcome competencies represented the domains of (1) public health, (4) organisation and management skills, and (5) professional and personal practice.

### 5.1. Input Competencies

#### 5.1.1. Role of the Preceptor

In terms of the preceptor’s role, the primary input competencies were related to (5.5) leadership and (6) education. Since the preceptors had to orientate and assist students to achieve their practice-based learning activities they (5.5.1.3) *led by example* by assuming the role of a mentor. The competency of leadership was further expanded when the preceptor identified the facility’s needs for the patient education poster, in-service training, and quality improvement project, thereby contributing to (5.5.2.3) *initiation, development and continuous improvement of pharmaceutical services.*

Since the provision of (6) education and training is integral to the preceptor’s role, it is more specifically aligned to (6.2.2.3) *shap[ing] and contribut[ing] to performance and learning needs of [students]*. Preceptors identified that the level of knowledge and readiness of third year students were sub-par and thus had to adjust their learning activities accordingly. In preparing the workplace or facility for the student rotation, preceptors had organised learning activities that were identified from learning assignment and outcomes which (6.3.1.3) *shape[d] and contribute[d] to the creation and development of the practice-based learning components* of the fourth year pharmacy curriculum. Participants noted the theoretical relevance of the clinical assignments with patients that often required them to arrange for clinical learning opportunities for students where they could sit in on doctor or nurse consultations.

“… I can see it’s making the right entrance because I thought that was it seriously and linking all that theory that they learnt in class to the … to practice here and especially the clinical side …” P25, interview, CHC 19, 2016.

#### 5.1.2. Student Assignments

Since PaCE students had more clinically oriented activities, the preceptor’s second role was to orientate the facility staff by communicating not only to the pharmacy personnel, but also to other healthcare professionals about the students’ learning activities. In addition, the interprofessional education assignment inherently required preceptors to develop connections with other healthcare professionals within their facilities, developing (5.7) collaborative practice networks. During 2016, participants advised their peers to build harmonious working relationships between the pharmacy and the rest of the facility to implement the programme more successfully.

“I think that for the PaCE program to work you need to have that relationship with the other professions as well and bring them in/draw them into it.” P21, interview, CHC 6, 2016.

This was especially evident in the CHCs, where preceptors had to create clinical experiences for students with clinical staff (in contrast to already existing ward rounds in hospitals) to enable students to contextualise their patient-centred assignments. In this way, preceptors (5.7.1.3) *advocate[d] for the inclusion of [pharmacy students] in multi-disciplinary teams*.

Three of the student assignments involved patient-centred activities, including the medicine history taking and reconciliation, care plan development and patient education poster. Student engagement was done either directly with the patient or indirectly via their medical records, which required the preceptor’s facilitation in (2) safe and rational use of medicines.

More specifically, for the medicine history and reconciliation student activity, preceptors had to (2.2.5.3) *create opportunities for the counselling and provision of information and advice to patients*. Some preceptors reported that by observing students conducting the activity at the pharmacy window, they were alerted to gaps in the patients’ pharmaceutical knowledge (even patients on chronic medication), which went unnoticed during routine dispensing at the window.

“… and those surveys [medicine history and reconciliations] … where they [the students] asked them [patients] about their medication … like do they know what it is used for … it was very interesting, because even people who have been on medication for a long time, still did not know …” P53, interview, CHC17, 2019.

“… and [students] asking all the set of questions with the format [medicine history and reconciliations] that you gave us. Because we [pharmacy staff at facility] never get to do that, we know we’re supposed to but you know how hectic it is.” P31, interview, CHC15, 2019.

The poster development activity specifically provided the opportunity for the preceptor to guide the students to (2.2.8.3) *develop [an] instructional aid to maximise patient counselling*. In addition, the in-service training assignment provided preceptors with the opportunity to (4.1.2.3) *identify staff training needs and facilitate appropriate training opportunities* through student-led activities. Since preceptors could also choose the audience for the training, most participants chose pharmacy personnel such as pharmacist’s assistants, but those participants who attended routine weekly clinical meetings at their sites, used this opportunity for students to educate all clinical facility staff.

“… we took them [the students] to the [weekly clinical] meeting just for them to observe. They also got a chance … to present to the doctors.” P46, interview, CHC13, 2019.

### 5.2. Output/Outcome Competencies

#### 5.2.1. Assignment Topics Identification from Facility-Based Needs

For the first two years of PaCE, preceptors were required to focus on patient education topics, relating to the pharmaceutical care aspects of asthma and diabetes for the patient education posters. The preceptor competencies that emerged were (1.6.2.3) *advocate[ing] for lifestyle changes that may prevent non-communicable diseases or improve the outcomes of medicine therapy*, and *(1.6.3.3) develop[ing] strategies to encourage patient to take responsibility for their own health and adherence to treatment guidelines*. However, since 2018 preceptors were encouraged to identify topics that were based on facility needs. The poster topics related to (1.1) promotion of health and wellness (1.2), medicine information, (1.6), primary healthcare and (1.7) pharmacovigilance.

In addition to choosing the topic for the patient education poster, the preceptors also had to choose the topics for the in-service training and quality improvement project based on their facility’s specific needs. The topics of the in-service trainings included mostly competencies clustered around the domains of (1) public health and (2) safe and rational use of medicines and medical devices.

The quality improvement (QI) project was often based on the medicine use evaluation (MUE) that the students had undertaken in third year SLiP. During PaCE, preceptors would instruct students to disseminate the outcomes of their facility’s MUE and offer prescribing recommendations that was aligned with standard treatment guidelines to prescribing staff. Preceptors were therefore able to (4.4.1.3) *contribute to regular audit activities, report and act upon findings* and (4.4.2.3) *use feedback from audits to improve services*. Since preceptors had the option to identify any topic (2019) from across domains 1 to 4 to improve pharmaceutical services, the QI student project topics contributed to (4.3.3.3) *develop and review workplace procedures and policies*, (4.3.4.3) *develop and review workflow systems in order to manage, prioritise and organise daily work and demonstrate time management skills* and (4.3.5.3) *ensure pharmaceutical infrastructure is in line with legislative requirements*.

“… you [the preceptor] identify things that are lacking, because they [the school nurses] just come and pick up the vaccines, nobody checks … that [the quality improvement project] opened space for us to do education with the school nurses and tell them what was needed and what are the requirements when you go out and do vaccine campaigns with the cooler box …” P49, workshop, CHC6, 2019.

#### 5.2.2. Emerging Competencies

Preceptors viewed the poster as a popular intervention to benefit the patients, as patient education was often lost during routine dispensing. Interestingly, participants identified a discord between practice and academic expectations. In this way they (5.1.1.3) *act[ed] as patient advocate[s] to ensure that patient care [was] optimised* in practice.

“So there is the academic expectation and the practical expectation and it doesn’t always [mix]. So you can end up getting a very good poster but ... the patient cannot benefit so much” P25, workshop, CHC19, 2019.

In addition, these patient-centred assignments, which required students to interact with other healthcare practitioners, resulted in the preceptors noticing how they were promoting the profession. In essence, such a convergence eased integration of pharmacy knowledge and expertise resulting in inter-professional collaborative practice between the students and other facility staff. Collaborative practice implicitly led to the (1.3.1.2) *promot[ion of] the role of the pharmacist in the healthcare team*.

“… nobody actually knows … how much does a pharmacist know.” P25, interview, CHC19, 2016.

Some preceptors took advantage of such an activity, using it as a resource to identify patient needs and had drawn on self-initiated interventions. These activities spoke to their professional practice competency of (5.1) patient-centred care.

“That [medicine history and reconciliations] also got me thinking, where are we lacking? … because there is no language barrier …” P48, interview, CHC15, 2019.

One of the self-management competencies that emerged as a consequence of the preceptor’s educational role was linked to (5.8.4.1) *identify[ing] gaps and areas for personal improvement and ensure its implementation*. Participants often referred to students as educational resources from whom they could retrieve new information, thereby motivating them to keep abreast with new knowledge.

“So basically … by doing this [preceptor] programme … it encourages us to keep current and updated.” P18, workshop1, CHC1, 2016.

“It’s a two way ... we [preceptors] learn, we grow, they [students] come in and they also test your level of knowledge ... I like it because it keeps you on your toes. And they ask you stuff and if you don’t know at that time you do your research and get back to them.” P39, workshop, CHC10, 2019.

## 6. Discussion

This paper is a synthesis from a longitudinal research project which explores pharmacists’ experiences as preceptors for the service learning programme of the School of Pharmacy at the UWC during 2016 (when the programme was expanded) and 2019. The stimulus for the sub-analysis of the data was the legal requirement imposed by the SAPC for pharmacists to complete CPDs for annual re-registration. By mapping competencies employed through precepting experiences, the School hopes to expand and adjust the presentation of the preceptor programme to facilitate pharmacists to more easily complete the new CPD requirements for pharmacists in South Africa, thereby strengthening the academic-service partnership.

The input competencies were those that every preceptor would employ while precepting students, thus was generically employed by all preceptors. The output competencies varied between preceptors based on the topics they chose according to the specific needs of their facilities, although emerging competencies were again uniform for all preceptors. However, caution should be used to generalise these competencies to all study participants because, variations in preceptor and facility preparedness for student rotations has been noted as a problem in literature [28].

The one competency that was both inherent and unique to being a preceptor included the provision of education and training and practice embedded education or workplace education. Preceptors in this study noted the difference in knowledge of students and had to design and implement educational activities based on their facility. Similarly, the competencies to determine the learner’s prior knowledge and skills in order to adapt activities and perform behaviours of designing placement activities to meet course goals and objectives have been recommended as essential for preceptors [29].

The competency of leadership was also associated with the preceptor role, specifically in terms of the behaviours of leading by example and driving continuous improvement of pharmaceutical services. These behaviours are related to the servant leadership style, which include characteristics such as listening, empathy, awareness, persuasion, conceptualization, foresight, stewardship, commitment to the growth of people and building community [30]. Leadership and role modelling have been identified in a Canadian study as a desired preceptor competency [31]. Similarly, the competency of “teach by example” has been noted as a key performance area of preceptors [29].

One of the self-management competencies that emerged for preceptors from their interactions with students was the identification of their own knowledge gaps and subsequently updating their knowledge. Similarly, 90% of the preceptors from a USA study agreed that being placed in preceptor roles had stimulated them and their staff to maintain their own knowledge and stay up-to-date with the current practices [32]. Indeed, being open to learn has been identified as a core competency for preceptors [29]. Furthermore, literature agrees that periodic refreshers for pharmacists on self-assessment according to competencies are important in promoting and motivating pharmacists to complete relevant CPDs [33].

The study participants mentioned expanding their collaborative practice networks at facility level on numerous occasions, including preparing the facility for students’ clinical experiences, an official assignment requiring interprofessional interaction, in some cases in-service training and the quality improvement project. Similarly, a study that evaluated the impact of interprofessional education (IPE) programmes in pharmacy education attested that developing practice sites to increase the opportunities for students to practice collaboratively was key [34]. The competency to create learning opportunities for students that highlight the roles and responsibilities of other healthcare professionals was essential to preceptors’ competencies [29]. This finding is supported by the FIP, who stated that inter-professional initiatives should begin before the student’s graduation and these interprofessional initiatives should persist through the course of their careers [35]. In addition, a recent consensus report on health professional education in South Africa endorsed transformative efforts aimed at inter-professional education and collaborative practice at the primary care level to optimise student learning outcomes [36]. Collaborative practice is especially relevant in the South African context as it is an essential component to enable the successful implementation of the NHI aimed at universal health coverage through PHC re-engineering. A core concept in PHC re-engineering is preventive care which is undertaken by multidisciplinary outreach teams [6].

The clinical focus of many of the student assignments stimulated preceptors to think in a patient-centred way because of the patient counselling and public health related competencies that were enabled during student rotations. The problem with these competencies so far is that, in most cases, preceptors primarily enable them through students and do not necessarily incorporate it into their own practice or the routine practices of the pharmacy. Similarly, a Croatian study that required community pharmacy preceptors to self-assess their own competencies found that preceptors rated their competencies related to public health and pharmaceutical care the lowest as compared to organization and management which they rated the highest [37]. A study from Qatar, that investigated self-assessment of preceptors, also showed that preceptors themselves scored lower for the patient care competency, as many hospital and clinic preceptors were involved in traditional dispensing [21]. Our results and the above two studies show that preceptors (from South Africa, Croatia and Qatar) generally tended to develop or were confident in competencies that they used routinely in practice. This may thus imply that routine pharmacy practice (which differs between countries) may be in part related to the competencies employed by preceptors.

In this study, the PaCE programme was seen as a platform for the preceptors to showcase pharmacy to doctors and nurses at their facility and show them the pharmacists’ knowledge and what they are capable of doing through clinical learning activities performed by pharmacy students in their facilities. This enabled the competency of professional and health advocacy. Indeed, promoting students to be included in ward rounds and team meetings has been allocated as an important performance area of preceptors [29]. Professional advocacy is important, as a Pakistani study noted that when pharmacy students participated in a clinical clerkship training programme at hospitals, the students were seen as an intrusion to the physicians and other allied healthcare professionals [38]. In fact, a South African study revealed that pharmacists were underutilised in the current health system in terms of clinical roles, because pharmacists tended to take responsibility for product-centred roles such as dispensing and procurement. Respondents suggested the implementation of the NHI could be used to create opportunities to utilise pharmacists’ scarce skills more effectively [39].

The identification of facility-specific needs by participants in this study and subsequently tailoring student assignments accordingly is an important behaviour that underlies the ability to do CPD. Literature has identified self-assessment, which is the initial step in identifying improvement needs as one of the steps of CPD that practitioners struggle with [33]. Another advantage from this study in terms of CPD competency is that a facility need is inherently practice-based, and bound to have an effect at the facility level. As such, preceptors employed the domain of organisation and management skills frequently in terms of the in-service training assignment as well as the quality improvement project. Although the in-service training was primarily knowledge-based, the quality improvement project did focus on quality assurance, which tended to focus on practice. Literature supports that student quality assurance projects based on the needs of the pharmacy has shown improved practice as reported by site preceptors [28]. Literature further recommends that successful models of CPD require coordination between quality improvement and an effective educational strategy [40].

The findings of this paper provide the first mapping of competencies employed by South African pharmacist preceptors. These findings can now add value to the reciprocity of the academic-service partnership to leverage the most impact. This is especially important in developing countries, because all resources should be maximized, and pharmacists’ engagement in practice-based CPD activities might be a start. The preceptor programme has two main advantages, namely developing pharmacist’s skills for CPD that can enhance pharmacy practice at the primary care level, and the adaptation of student assignments to increase the importance of facility needs and quality improvements to strengthen the health system. Additionally, taking advantage of policies like NHI and PHC re-engineering provides a window of opportunity for educators at UWC to advocate for the inclusion of the pharmacist in the multidisciplinary team, while utilising the mandatory CPD implementation driven by the SAPC. In addition, demonstrating competency in public health is especially valuable in the current South African policy context. Integrating this according to a health system strengthening approach [41] might allow for current and future pharmacists to enhance their role for pharmaceutical care and the prioritization of patient-centred care. The triple global public health drivers—the sustainable development goals, universal health coverage and primary healthcare—offer educators excellent cohesive and synergising opportunities, as shown in this case study.

## 7. Limitations

One of the limitations of this study includes that competencies mapped for preceptors might be an overestimation for some preceptors that did not engage very efficiently with the precepting functions due to varying factors, such as workload and limited time spent with students. In addition, these results are mainly based on self-report by the preceptor and might not depict their actual level of performance during student supervision. In addition, no objective student assessment of the preceptors’ performance was undertaken. Lastly, these competencies were mapped for pharmacists working in public sector pharmacies in South Africa, where pharmacists are not routinely involved in direct patient care or clinical activities. The competencies will thus not be generalizable to practice environments were pharmaceutical care is routinely practiced.

## 8. Conclusions

The preceptor programme enabled pharmacist preceptors to employ several competencies that are aligned with the SAPC’s competency framework. The findings from this study revealed that competencies inherent in the role of the preceptor included those from education, leadership, patient counselling, collaborative practice and human resource management. The facility needs identification assignments that provided the preceptors with the freedom to develop practice-based competencies were associated with improving pharmaceutical infrastructure, quality assurance, professional and health advocacy, primary healthcare, self-management and patient-centred care. These findings show the potential for the practice-based preceptor programme to steer the CPD of pharmacist preceptors towards improved individual performance and health outcomes through emphasizing a quality improvement approach.

## Figures and Tables

**Figure 1 pharmacy-08-00096-f001:**
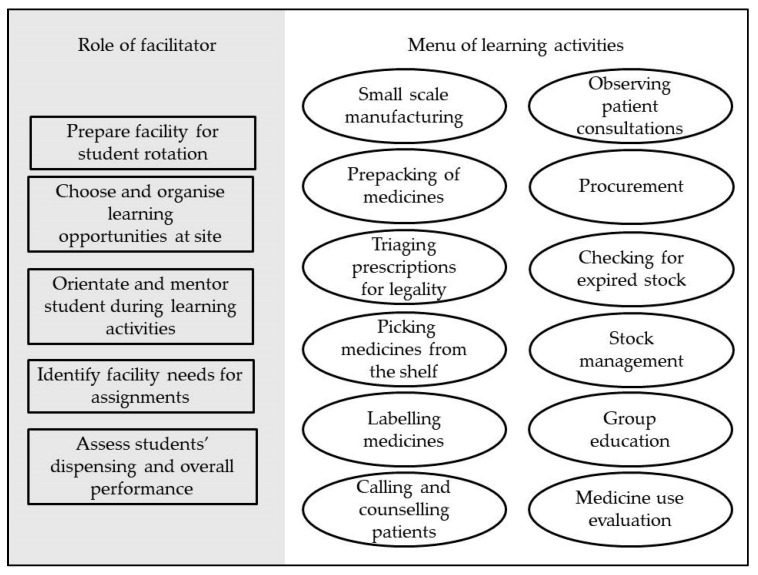
Pharmacist facilitators’ roles and menu of student learning activities for the third-year service-learning programme (pre-2013 curriculum).

**Figure 2 pharmacy-08-00096-f002:**
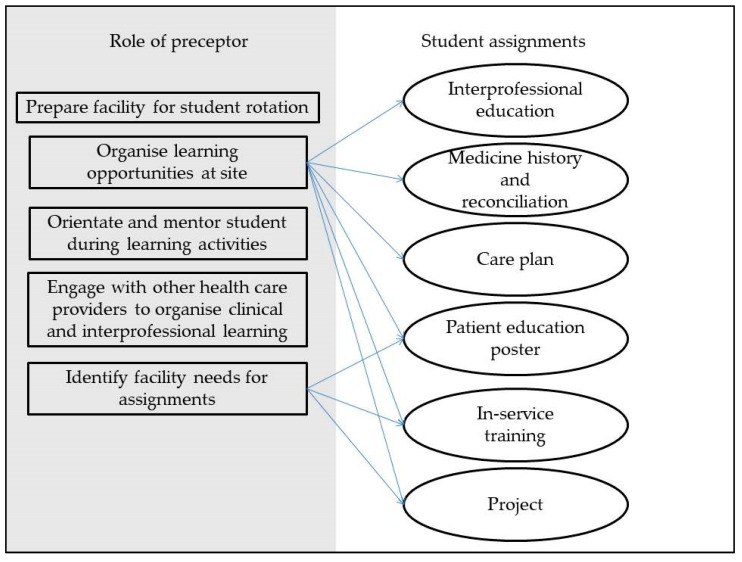
Pharmacist preceptors’ roles and learning assignments for the fourth year clinical programme (post-2013 curriculum).

**Figure 3 pharmacy-08-00096-f003:**
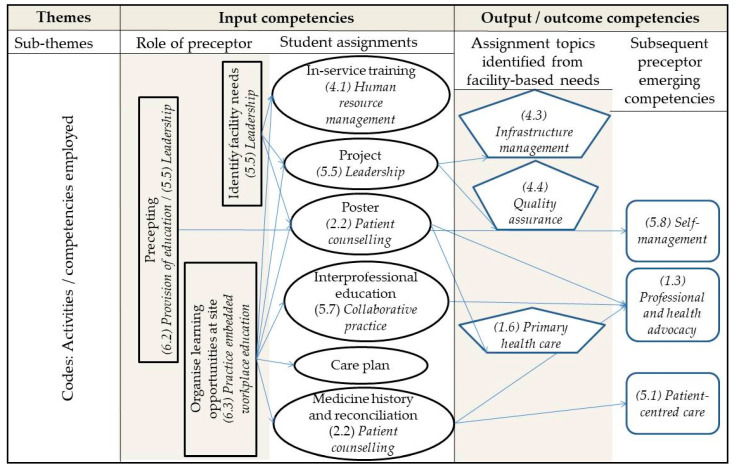
Illustration of themes and sub-themes in relation to competencies (*in italics*) employed by pharmacist preceptors during student rotations.

**Table 1 pharmacy-08-00096-t001:** Summary of domains and competencies for South African pharmacists (2017) [22].

Domains (Competency Clusters)	Competencies
1. Public Health	1.1 Promotion of health and wellness 1.2 Medicines information 1.3 Professional and health advocacy 1.4 Health economics 1.5 Epidemic and disaster management 1.6 Primary healthcare 1.7 Pharmacovigilance
2. Safe and rational use of medicines and medical devices	2.1 Patient consultation skills 2.2 Patient counselling skills 2.3 Patient medicine review and management 2.4 Medicines and medical devices safety 2.5 Therapeutic outcome monitoring 2.6 Pharmacist initiated therapy
3. Supply of medicines	3.1 Clinical trials 3.2 Medicine production according to GxP 3.3 Supply chain management 3.4 Formulary development 3.5 Medicine dispensing 3.6 Medicine compounding 3.7 Medicine disposal/destruction
4. Organisation and management skills	4.1 Human resources management 4.2 Financial management 4.3 Facility and infrastructure management 4.4 Quality assurance 4.5 Change management 4.6 Policy development
5. Professional and personal practice	5.1 Patient-centred care 5.2 Professionally practice 5.3 Ethical and legal practice 5.4 Continuing Professional Development 5.5 Leadership skills 5.6 Decision-making skills 5.7 Collaborative practice 5.8 Self-management skills 5.9 Communication skills
6. Education, Critical analyses, and Research	6.1 Education and training policy 6.2 Provision of education and training 6.3 Practice embedded education or workplace education 6.4 Gap analysis 6.5 Critical analysis 6.6 Research 6.7 Supervision of other researchers 6.8 Collaborative research

Key: GxP = Good manufacturing practice.

**Table 2 pharmacy-08-00096-t002:** Framework of experiential learning programmes and service partners at the University of the Western Cape (UWC) School of Pharmacy pre and post 2013.

Pre-2013 Curriculum	Post-2013 Curriculum
1st year: no experiential learning programme	2013: 1st year—focuses on health and wellness and environmental health within the community (service partners: primary schools, children’s homes, non-governmental organizations, City of Cape Town—environmental health services)
2nd year: no experiential learning programme	2014: 2nd year—focuses on preventative and primary healthcare services (service partner: City of Cape Town)
2009–2014: 3rd year—primary healthcare group project (service partner: City of Cape Town)	2015: 3rd year (SLiP)—focuses on pharmaceutical service delivery (2 weeks) (service partner: MDHS)
2002–2015: 4th year (SLiP)—focuses on pharmaceutical service delivery (3 weeks), and clinical blocks (2–3 days) (service partner: MDHS)	2016: 4th year (PaCE)—focuses on direct patient clinical assessments and pharmaceutical interventions (10 weeks) (service partner: MDHS)

Key: SLiP: service learning in pharmacy, MDHS: Metro District Health Services, PaCE: Patient Care Experience.

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
