# Peer review of "Continuous Professional Development for Public Sector Pharmacists in South Africa: A Case Study of Mapping Competencies in a Pharmacists’ Preceptor Programme"

_pharmacy, 2020, doi:10.3390/pharmacy8020096_

Round 1
Reviewer 1 Report
Thank your for the opportunity to review this case report. I enjoyed reading the well written paper and have made some minor suggestions below.
Line 173 - Figure 1 - this Figure discusses the Service learning program for third year students. Could the authors please confirm if this is for the pre-2013 or post-2013 curriculum? The paragraphs around this figure prefer to the pre-2013 curriculum and this causes confusion.
Line 200- It would be good to include that this figure refers to the post 2013 curriculum
Line 203 - I would suggest that the authors make a separate statement about the ethics approvals. While the approvals have been added, I feel that is would be preferred if the approvals are described in a discrete sentence.
Line 218 - I would suggest that the authors consider modifying the results of the qualitative interviews into themes and use sub-heading to focus the reader on specific outcomes.
Line 221 - could the authors please describe "sub-structure management" pharmacists mean?
Line 450 - I would like the limitations put under a subheading "limitations".
It may be useful for the authors to add a link to the SAPC’s competency framework in the conclusion to readers can easily find the information.
Author Response
Response to Reviewer 1 Comments
We would like to thank reviewer 1 for the constructive comments that assisted us in improving the readability of this paper.
Point 1: Line 173 - Figure 1 - this Figure discusses the Service learning program for third year students. Could the authors please confirm if this is for the pre-2013 or post-2013 curriculum? The paragraphs around this figure prefer to the pre-2013 curriculum and this causes confusion.
Response 1: It was referring to the pre-2013 curriculum, changed figure caption as follows - Figure 1. Pharmacist facilitators’ roles and menu of student learning activities for the third year service learning programme (pre-2013 curriculum).
Point 2: Line 200- It would be good to include that this figure refers to the post 2013 curriculum
Response 2: Included as follows - Figure 2. Pharmacist preceptors’ roles and learning assignments for the fourth year clinical programme (post-2013 curriculum).
Point 3: Line 203 - I would suggest that the authors make a separate statement about the ethics approvals. While the approvals have been added, I feel that is would be preferred if the approvals are described in a discrete sentence.
Response 3: added the sentence - Ethics approvals were obtained from the UWC Senate Research Ethics Committee (15/6/95) and from UWC Biomedical Research Ethics Committee (BM18/6/14).
Point 4: Line 218 - I would suggest that the authors consider modifying the results of the qualitative interviews into themes and use sub-heading to focus the reader on specific outcomes.
Response 4: Figure 3 was modified to illustrate the themes and sub-themes. The results text narrative was re-arranged according to sub-headings that better guides the reader to follow the mapped competencies in terms of the themes and sub-themes in a more structured manner.
Point 5: Line 221 - could the authors please describe "sub-structure management" pharmacists mean?
Response 5: South Africa has a district-based health system i.e. each province is divided into districts and subsequently smaller sub-districts. Cape Town metropole health district is divided into 8 sub-districts of which the management structure is divided into four sub-structures (i.e. each sub-structure are constituted by two sub-districts). Because sub-district is a more reader friendly term, we have changed all references to sub-structure to sub-district to increase clarity for the reader.
Point 6: Line 450 - I would like the limitations put under a subheading "limitations".
Response 6: added heading
Point 7: It may be useful for the authors to add a link to the SAPC’s competency framework in the conclusion to readers can easily find the information.
Response 7: Added the link at reference 22 Competency Standards for Pharmacists in South Africa:
https://www.mm3admin.co.za/documents/docmanager/0C43CA52-121E-4F58-B8F6-81F656F2FD17/00126360.pdf
Reviewer 2 Report
This is a very interesting and unique look at pharmacy in another country and the steps that are being taken to advance the profession.
There was a lot of information presented in the article - I am slightly confused regarding the relationship between the CPD and the preceptor program - were preceptors provided CPD credit after participating in the SLiP program or simply as a method to "nudge" them to complete their requirements? If it is the latter, is there data to suggest that participation as a preceptor worked to facilitate future completion of CPD?
Also, while I understand the need to discuss the competencies utilized within the SLiP program, it also is a bit confusing as to why it was included - what does this have to do with them completing their CPD requirements?
In line 50, I would encourage that you add some clarification when you state "6 CPD" - perhaps add the word events?
Furthermore, in discussion of the interview results, you list many of the quotes in text; my understanding is that the gold standard is to summarize the results in the text and include the actual quotes in a table organized by theme for easier reading.
Author Response
v
Response to Reviewer 2 Comments
We would like to that reviewer 2 in assisting us in improving the readability of this paper for readers from different contexts.
Point 1: There was a lot of information presented in the article - I am slightly confused regarding the relationship between the CPD and the preceptor program - were preceptors provided CPD credit after participating in the SLiP program * or simply as a method to "nudge" them to complete their requirements **? If it is the latter, is there data to suggest that participation as a preceptor worked to facilitate future completion of CPD?
Response 1: (*no, **yes, but it is only proposed at this point) In terms of answering the latter question, to our knowledge and effort, there does not seem to be literature from preceptor programmes that guide and support preceptors to complete their CPDs entries. Refer to lines 89-93; 362-365. Since the pharmacists were already undertaking the responsibilities to meet the students learning activities well before SAPC’s mandatory CPD requirements (2019), this article unveils how their preceptor roles could be translated towards matching the pool of competencies required for CPD submissions for 2020 and beyond. We have thus reworded lines 89-93 to: “This paper uses a preceptor programme (linked to the University of the Western Cape School of Pharmacy’s service learning programme) to (1) identify competencies employed by preceptors (in primary care public health care facilities in Cape Town), and (2) map these competencies according to the new competency standards for pharmacists in South Africa. This information will be used to shape future engagement between university faculty and preceptors to increase the synergy between preceptors, faculty and students in participating in the programme i.e. strengthening the reciprocity of the academic-service partnership, by aligning with a mandatory activity required by the SAPC as part of the programme.”
Point 2: Also, while I understand the need to discuss the competencies utilized within the SLiP program, it also is a bit confusing as to why it was included - what does this have to do with them completing their CPD requirements?
Response 2: Currently, the preceptor programme is not linked to the CPD requirements, it is two separate (additional) activities pharmacists have to complete above and beyond their routine work commitments. As CPD is new for pharmacists and will require more time commitments from preceptors, by aligning the preceptor programme with CPD, we might ease some of the effort and increase the benefits of participating in the preceptor programme (as public sector pharmacists are generally already overloaded). By aligning these two activities (precepting and CPD) it works towards a win-win scenario of pharmacy practice and pharmacy education by maximising available resources. The revision as indicated in point 1 might clarify this.
Point 3: In line 50, I would encourage that you add some clarification when you state "6 CPD" - perhaps add the word events?
Response 3: Changed to: “6 CPD entries”
Point 4: Furthermore, in discussion of the interview results, you list many of the quotes in text; my understanding is that the gold standard is to summarize the results in the text and include the actual quotes in a table organized by theme for easier reading.
Response 4: As reviewer 1 also suggested ways to improve the readability of the results section, we opted for revisions that align to the suggestions of both reviewers. As such, Figure 3 was modified to illustrate the themes and sub-themes of the results. In addition, the results text narrative was re-arranged according to sub-headings that better guides the reader to follow the mapped competencies in terms of the themes and sub-themes in a more structured manner. The quotes in the narrative text have been retained, because it seems to facilitate connection more explicitly with the behavioural statements.